# Multi-armed Bandits with Compensation

**Siwei Wang**
IIIS, Tsinghua University
wangsw15@mails.tsinghua.edu.cn

**Longbo Huang**[*]
IIIS, Tsinghua University
longbohuang@tsinghua.edu.cn

## Abstract

We propose and study the known-compensation multi-armed bandit (KCMAB) problem, where a system controller offers a set of arms to many short-term players for $T$ steps. In each step, one short-term player arrives at the system. Upon arrival, the player aims to select an arm with the current best average reward and receives a stochastic reward associated with the arm. In order to incentivize players to explore other arms, the controller provide proper payment compensations to players. The objective of the controller is to maximize the total reward collected by players while minimizing the total compensation. We first provide a compensation lower bound $\Theta(\sum_i \frac{\Delta_i \log T}{KL_i})$, where $\Delta_i$ and $KL_i$ are the expected reward gap and the Kullback-Leibler (KL) divergence between distributions of arm $i$ and the best arm, respectively. We then analyze three algorithms for solving the KCMAB problem, and obtain their regrets and compensations. We show that the algorithms all achieve $O(\log T)$ regret and $O(\log T)$ compensation that match the theoretical lower bounds. Finally, we present experimental results to demonstrate the performance of the algorithms.

## 1 Introduction

Multi-armed bandit (MAB) is a game that lasts for an unknown time horizon $T$ [4, 17]. In each time slot, the controller pulls one out of $N$ arms, and pulling different arms results in different feedbacks. In the stochastic MAB model [12], feedbacks from each single arm follow a corresponding distribution, which is unknown to the controller. These feedbacks are random variables independent of any other events. After pulling the arm, the controller collects a reward that depends on the feedback. The controller aims to maximize the sum of rewards during the game by choosing a proper arm to pull in each time slot, and the decision can depend on all available information, i.e., past chosen arms and feedbacks. The common metric for evaluating the performance of a policy is the value of regret, defined as the expected difference between the controller's reward and pulling an arm that generates the largest expected reward.

The MAB formulation models the trade-off between exploration and exploitation, where exploration concerns finding the potential best arms, but can result in pulling sub-optimal arms, while exploitation aims at choosing arms with the current best performance and can lose reward if that arm is in fact sub-optimal. Thus, optimizing this trade-off is very important for any controller seeking to minimize regret. However, in many real-world applications, arms are not pulled by the controller concerning long-term performance. Instead, actions are taken by short-term players interested in optimizing their instantaneous reward. In this case, an important means is to provide monetary compensations to players, so that they act as if they are pulling arms on behalf of the controller, to jointly minimize regret, e.g., [8]. This is precisely our focus in this paper, i.e., we aim to design an efficient incentivizing policy, so as to minimize regret while not giving away too much compensation.

---

[*]This work is supported in part by the National Natural Science Foundation of China Grants 61672316, 61303195, the Tsinghua Initiative Research Grant, and the China Youth 1000-Talent Grant

As a concrete example, consider the scenario where an e-commerce website recommends goods to consumers. When a consumer chooses to purchase a certain good, he receives the reward of that good. The website similarly collects the same reward as a recognition of the recommendation quality. In this model, the website acts as a controller that decides how to provide recommendations. Yet, the actual product selection is made by consumers, who are not interested in exploration and will choose to optimize their reward greedily. However, being a long-term player, the website cares more about maximizing the total reward throughout the game. As a result, he needs to devise a scheme to influence the choices of short-term consumers, so that both the consumers and website can maximize their benefits. One common way to achieve this goal in practice is that the website offers customized discounts for certain goods to consumers, i.e., by offering a compensation to pay for part of the goods. In this case, each customer, upon arrival, will choose the good with largest expected reward plus the compensation. The goal of the e-commerce site is to find an optimal compensation policy to minimize his regret, while not spending too much additional payment.

It is important to notice the difference between regret and compensation. In particular, regret comes from pulling a sub-optimal arm, while compensation comes from pulling an arm with poor past behavior. For example, consider two arms with expected rewards $0.9$ for arm 1 and $0.1$ for arm 2. Suppose in the first twenty observations, arm 1 has an empirical mean $0.1$ but arm 2 has an empirical mean $0.9$. Then, in the next time slot, pulling arm 2 will cause regret $0.8$, since its expected gain is $0.8$ less than arm 1. But in a short-term player's view, arm 2 behaves better than arm 1. Thus, pulling arm 2 does not require any compensation, while pulling arm 1 needs $0.8$ for compensation. As a result, the two measures can behave differently and require different analysis, i.e., regret depends heavily on learning the arms well, while compensation is largely affected by how the reward dynamics behaves.

There is a natural trade-off between regret and compensation. If one does not offer any compensation, the resulting user selection policy is greedy, which will lead to a $\Theta(T)$ regret. On the other hand, if one is allowed to have arbitrary compensation, one can achieve an $O(\log T)$ regret with many existing algorithms. The key challenge in obtaining the best trade-off between regret and compensation lies in that the compensation value depends on the random history. As a consequence, different random history not only leads to different compensation value, but also results in different arm selection. Moreover, in practice, the compensation budget may be limited, e.g., a company hopes to maximize its total income which equals to reward subtracts compensation. These make it hard to analyze its behavior.

## 1.1 Related works

The incentivized learning model has been investigated in prior works, e.g., [8, 14, 15]. In [8], the model contains a prior distribution for each arm's mean reward at the beginning. As time goes on, observations from each arm update the posterior distributions, and subsequent decisions are made based on posterior distributions. The objective is to optimize the total discounted rewards. Following their work, [14] considered the case when the rewards are not discounted, and they presented an algorithm to achieve regret upper bound of $O(\sqrt{T})$. In [15], instead of a simple game, there is a complex game in each time slot that contains more players and actions. These incentivization formulations can model many practical applications, including crowdsourcing and recommendation systems [6, 16].

In this paper, we focus on the non-Bayesian setting and consider non-discounted rewards. As pointed out in [14], the definition of user expectation is different in this case. Specifically, in our setting, each player selects arms based on their empirical means, whereas in the Bayesian setting, it is possible for a player to also consider posterior distributions of arms for decision making. We propose three algorithms for solving our problem, which adapt ideas from existing policies for stochastic MAB, i.e., Upper Confidence Bound (UCB) [2, 9], Thompson Sampling (TS) [18] and $\varepsilon$-greedy [20]. These algorithms guarantee $O(\log T)$ regret upper bounds (match the regret lower bound $\Theta(\log T)$ [12]).

Another related bandit model is contextual bandit, where a context is contained in each time slot [3, 5, 13]. The context is given before a decision is made, and the reward depends on the context. As a result, arm selection also depends on the given context. In incentivized learning, the short-term players can view the compensation as a context, and their decisions are influenced by the context. However, different from contextual bandits, where the context is often exogenous and the controller focuses on identifying the best arm under given contexts, in our case, the context is given by the controller and itself is influenced by player actions. Moreover, the controller needs to pay for

obtaining a desired context. What he needs is the best way to construct a context in every time slot, so that the total cost is minimized.

In the budgeted MAB model, e.g., [7, 19, 21], players also need to pay for pulling arms. In this model, pulling each arm costs a certain budget. The goal for budgeted MAB is to maximize the total reward subject to the budget constraint. The main difference from our work is that in budgeted MAB, the cost budget for pulling each arm is pre-determined and it does not change with the reward history. In incentivized learning, however, different reward sample paths will lead to different costs for pulling the same arm.

## 1.2 Our contributions

The main contributions of our paper are summarized as follows:

1. We propose and study the Known-Compensation MAB problem (KCMAB). In KCMAB, a long-term controller aims to optimize the accumulated reward but has to offer compensation to a set of short-term players for pulling arms. Short-term players, on the other hand, arrive at the system and make greedy decisions to maximize their expected reward plus compensation. The objective of the long-term controller is to design a proper compensation policy, so as to minimize his regret with minimum compensation. KCMAB is a non-Bayesian and non-discounted extension of the model in [8].

2. In KCMAB, subject to the algorithm having an $o(T^{\alpha})$ regret for any $\alpha \in (0, 1)$, we provide a $\Theta(\log T)$ lower bound for the compensation. This compensation lower bound has the same order as the regret lower bound, which means that one cannot expect a compensation to be much less than its regret, if the regret is already small.

3. We propose algorithms to solve the KCMAB problem and present their compensation analysis. Specifically, we provide the analyses of compensation for the UCB policy, a modified $\varepsilon$-greedy policy and a modified-TS policy. All these algorithms have $O(\log T)$ regret upper bounds while using compensations upper bounded $O(\log T)$, which matches the lower bound (in order).

4. We provide experimental results to demonstrate the performance of our algorithms. In experiments, we find that the modified TS policy behaves better than UCB policy, while the modified $\varepsilon$-greedy policy has regret and compensation slightly larger than those under the modified-TS policy. We also compare the classic TS algorithm and our modified-TS policy. The results show that our modification is not only effective in analysis, but also impactful on actual performance. Our results also demonstrate the trade-off between regret and compensation.

## 2 Model and notations

In the Known-Compensation Multi-Armed Bandit (KCMAB) problem, a central controller has $N$ arms $\{1, \cdots, N\}$. Each arm $i$ has a reward distribution denoted by $D_i$ with support $[0, 1]$ and mean $\mu_i$. Without loss of generality, we assume $1 \geq \mu_1 > \mu_2 \geq \cdots \mu_N \geq 0$ and set $\Delta_i = \mu_1 - \mu_i$ for all $i \geq 2$. The game is played for $T$ time steps. In each time slot $t$, a short-term player arrives at the system and chooses an arm $a(t)$ to pull. After the player pulls arm $a(t)$, the player and the controller each receive a reward drawn from the distribution $D_{a(t)}$, denoted by $X_{a(t)}(t) \sim D_{a(t)}$, which is an independent random variable every time arm $a(t)$ is pulled.

Different from the classic MAB model, e.g., [12], where the only control decision is arm selection, the controller can also choose to offer a compensation to a player for pulling a particular arm, so as to incentivize the player to explore an arm favored by the controller. We denote the offered compensation by $c(t) = c_{a(t)}(t)$, and assume that it can depend on all the previous information, i.e., it depends on $\mathcal{F}_{t-1} = \{(a(\tau), X(\tau), c(\tau)) | 1 \leq \tau \leq t - 1\}$. Each player, if he pulls arm $i$ at time $t$, collects income $\hat{\mu}_i(t) + c_i(t)$, where $\hat{\mu}_i(t) \triangleq M_i(t)/N_i(t)$ is the empirical mean reward of arm $i$, with $N_i(t) = \sum_{\tau < t} \mathbb{I}[a(\tau) = i]$ being the total number of times for pulling arm $i$ and $M_i(t) = \sum_{\tau < t} \mathbb{I}[a(\tau) = i] X(t)$ being the total reward collected. Each player is assumed to greedily choose the arm $i = \text{argmax}_j \{\hat{\mu}_j(t) + c_j(t)\}$.

The long-term controller, on the other hand, concerns about the expected total reward. Following the MAB tradition, we define the following total regret:

$$Reg(T) = T \max_i \mu_i - Rew(T) = T\mu_1 - Rew(T),$$

where $Rew(T)$ denotes the expected total reward that the long-term controller can obtain until time horizon $T$. We then use $Com_i(T) = \mathbb{E}\left[\sum_{\tau=1}^{T} \mathbb{I}[a(\tau) = i]c(\tau)\right]$ to denote the expected compensation paid for arm $i$, and denote $Com(T) = \sum_i Com_i(T)$ the expected total compensation.

It is known from [12] that $Reg(T)$ has a lower bound of $\Omega(\sum_{i=2}^{N} \frac{\Delta_i \log T}{KL(D_i,D_1)})$, where $KL(P,Q)$ denotes the Kullback-Leibler (KL)-divergence between distributions $P$ and $Q$, even when a single controller is pulling arms for all time. Thus, our objective is to minimize the compensation while keeping the regret upper bounded by $O(\sum_{i=2}^{N} \frac{\Delta_i \log T}{KL(D_i,D_1)})$.

Note that in the non-Bayesian model, if there are no observations for some arm $i$, players will have zero knowledge about its mean reward and they cannot make decisions. Thus, we assume without loss of generality that in the first $N$ time slots of the game, with some constant compensation, the long-term controller can control the short-term players to choose all the arms once. This assumption does not influence the results in this paper.

In the following, we will present our algorithms and analysis. Due to space limitation, all complete proofs in this paper are deferred to the supplementary file. We only provide proof sketches in the main text.

## 3   Compensation lower bound

In this section, we first derive a compensation lower bound, subject to the constraint that the algorithm guarantees an $o(T^\alpha)$ regret for any $\alpha \in (0, 1)$. We will make use of the following simple fact to simplify the computation of compensation at every time slot.

**Fact 1.** *If the long-term controller wants a short-term player to choose arm $i$ in time slot $t$, the minimum compensation he needs to pay is $c_i(t) = \max_j \hat{\mu}_j(t) - \hat{\mu}_i(t)$.*

With Fact 1, we only need to consider the case $c_i(t) = \max_j \hat{\mu}_j(t) - \hat{\mu}_i(t)$ for each arm $i$.

**Theorem 1.** *In KCMAB, if an algorithm guarantees an $o(T^\alpha)$ regret upper bound for any fixed $T$ and any $\alpha \in (0, 1)$, then there exist examples of Bernoulli Bandits, i.e., arms having reward $0$ or $1$ every time, such that the algorithm must pay $\Omega\left(\sum_{i=2}^{N} \frac{\Delta_i \log T}{KL(D_i,D_1)}\right)$ for compensation in these examples.*

**Proof Sketch**: Suppose an algorithm achieves an $o(T^\alpha)$ regret upper bound for any $\alpha \in (0, 1)$. We know that it must pull a sub-optimal arm $i$ for $\Omega\left(\frac{\log T}{KL(D_i,D_1)}\right)$ times almost surely [12]. Now denote $t_i(k)$ the time slot (a random variable) that we choose arm $i$ for the $k$-th time. We see that one needs to pay $\mathbb{E}[\max_j \hat{\mu}_j(t_i(k)) - \hat{\mu}_i(t_i(k))] \geq \mathbb{E}[\hat{\mu}_1(t_i(k))] - \mathbb{E}[\hat{\mu}_i(t_i(k))]$ for compensation in that time slot. By definition of $t_i(k)$ and the fact that all rewards are independent with each other, we always have $\mathbb{E}[\hat{\mu}_i(t_i(k))] = \mu_i$.

It remains to bound the value $\mathbb{E}[\hat{\mu}_1(t_i(k))]$. Intuitively, when $\mu_1$ is large, $\mathbb{E}[\hat{\mu}_1(t_i(k))]$ cannot be small, since those random variables are with mean $\mu_1$. Indeed, when $\mu_1 > 0.9$ and $D_1$ is a Bernoulli distribution, one can prove that $\mathbb{E}[\hat{\mu}_1(t_i(k))] \geq \frac{\mu_1}{2} - 2\delta(T)$ with a probabilistic argument, where $\delta(T)$ converges to 0 as $T$ goes to infinity. Thus, for large $\mu_1$ and small $\mu_2$ (so are $\mu_i$ for $i \geq 2$), we have that $\mathbb{E}[\hat{\mu}_1(t_i(k))] - \mu_i = \Omega(\mu_1 - \mu_i)$ holds for any $i$ and $k \geq 2$. This means that the compensation we need to pay for pulling arm $i$ once is about $\Theta(\mu_1 - \mu_i) = \Omega(\Delta_i)$. Thus, the total compensation $\Omega\left(\sum_{i=2}^{N} \frac{\Delta_i \log T}{KL(D_i,D_1)}\right)$. $\square$

## 4   Compensation upper bound

In this section, we propose three algorithms that can be applied to solve the KCMAB problem and present their analyses. Specifically, we consider the Upper Confidence Bound (UCB) Policy [2], and

```
1: for t = 1, 2, · · · , N do
2:     Choose arm a(t) = t.
3: end for
4: for t = N + 1, · · · do
5:     For all arm i, compute r_i(t) = √(2 log t / N_i(t)) and u_i(t) = μ̂_i(t) + r_i(t)
6:     Choose arm a(t) = argmax_i u_i(t) (with compensation max_j μ̂_j(t) − μ̂_{a(t)}(t))
7: end for
```

**Algorithm 1:** The UCB algorithm for KCMAB.

propose a modified $\varepsilon$-Greedy Policy and a modified-Thompson Sampling Policy. Note that while the algorithms have been extensively analyzed for their regret performance, the compensation metric is significantly different from regret. Thus, the analyses are different and require new arguments.

### 4.1 The Upper Confidence Bound policy

We start with the UCB policy shown in Algorithm 1. In the view of the long-term controller, Algorithm 1 is the same as the UCB policy in [2], and its regret has been proven to be $O\left(\sum_{i=2}^{N} \frac{\log T}{\Delta_i}\right)$. Thus, we focus on the compensation upper bound, which is shown in Theorem 2.

**Theorem 2.** *In Algorithm 1, we have that*

$$Com(T) \leq \sum_{i=2}^{N} \frac{16 \log T}{\Delta_i} + \frac{2N\pi^2}{3}$$

**Proof Sketch**: First of all, it can be shown that the each sub-optimal arm is pulled for at most $\frac{8}{\Delta_i^2} \log T$ times in Algorithm 1 with high probability. Since in every time slot $t$ the long-term controller chooses the arm $a(t) = \text{argmax}_j \hat{\mu}_j(t) + r_j(t)$, we must have $\hat{\mu}_{a(t)}(t) + r_{a(t)}(t) = \max_j(\hat{\mu}_j(t) + r_j(t)) \geq \max_j \hat{\mu}_j(t)$. This implies that the compensation is at most $r_{a(t)}(t)$. Moreover, if arm $a(t)$ has been pulled the maximum number of times, i.e., $N_{a(t)}(t) = \max_j N_j(t)$, then $r_{a(t)}(t) = \min_j r_j(t)$ (by definition). Thus, $\hat{\mu}_{a(t)}(t) = \max_j \hat{\mu}_j(t)$, which means that the controller does not need to pay any compensation.

Next, for any sub-optimal arm $i$, with high probability, the compensation that the long-term controller pays for it can be upper bounded by:

$$Com_i(T) \leq \mathbb{E}\left[\sum_{\tau=1}^{N_i(T)} \sqrt{\frac{2 \log T}{\tau}}\right] \leq \mathbb{E}\left[\sqrt{8 N_i(T) \log T}\right] \overset{(a)}{\leq} \sqrt{8 \mathbb{E}[N_i(T)] \log T} \leq \frac{8 \log T}{\Delta_i}$$

Here the inequality (a) holds because $\sqrt{x}$ is concave. As for the optimal arm, when $N_1(t) \geq \sum_{i=2}^{N} \frac{8 \log T}{\Delta_i^2}$, with high probability $N_1(t) = \max_j N_j(t)$. Thus, the controller does not need to pay compensation in time slots with $a(t) = 1$ and $N_1(t) \geq \sum_{i=2}^{N} \frac{8 \log T}{\Delta_i^2}$. Using the same argument, the compensation for arm 1 is upper bounded by $Com_1(T) \leq \sum_{i=2}^{N} \frac{8 \log T}{\Delta_i}$ with high probability. Therefore, the overall compensation upper bound is given by $Com(T) \leq \sum_{i=2}^{N} \frac{16 \log T}{\Delta_i}$ with high probability. □

### 4.2 The modified $\varepsilon$-greedy policy

The second algorithm we propose is a modified $\varepsilon$-greedy policy, whose details are presented in Algorithm 2. The modified $\varepsilon$-greedy algorithm, though appears to be similar to the classic $\varepsilon$-greedy algorithm, has a critical difference. In particular, instead of randomly choosing an arm to explore, we use the round robin method to explore the arms. This guarantees that, given the number of total explorations, each arm will be explored a deterministic number of times. This facilitates the analysis for compensation upper bound.

In the regret analysis of the $\varepsilon$-greedy algorithm, the random exploration ensures that at time slot $t$, the expectation of explorations on each arm is about $\frac{\epsilon}{N} \log t$. Thus, the probability that its empirical

```
1: Input: ε,
2: for t = 1, 2, · · · , N do
3:     Choose arm a(t) = t.
4: end for
5: a_e ← 1
6: for t = N + 1, · · · do
7:     With probability min{1, ε/t}, choose arm a(t) = a_e and set a_e ← (a_e  mod N) + 1 (with
        compensation max_j μ̂_j(t) − μ̂_{a(t)}(t)).
8:     Else, choose the arm a(t) = argmax_i μ̂_i(t).
9: end for
```

**Algorithm 2:** The modified $\varepsilon$-greedy algorithm for KCMAB.

mean has a large error is small. In our algorithm, the number of explorations of each single arm is almost the same as classic $\varepsilon$-greedy algorithm in expectation (with only a small constant difference). Hence, adapting the analysis from the $\varepsilon$-greedy algorithm gives the same regret upper bound, i.e. $O(\sum_{i=2}^{N} \frac{\Delta_i \log T}{\Delta_2^2})$ when $\epsilon = \frac{cN}{\Delta_2^2}$.

Next, we provide a compensation upper bound for our modified $\varepsilon$-greedy algorithm.

**Theorem 3.** *In Algorithm 2, if we have $\epsilon = \frac{cN}{\Delta_2^2}$, then*

$$Com(T) \leq \sum_{i=2}^{N} \frac{c\Delta_i \log T}{\Delta_2^2} + \frac{N^2}{2\Delta_2}\sqrt{c \log T}. \tag{1}$$

**Proof Sketch:** Firstly, our modified $\varepsilon$-greedy algorithm chooses the arm with the largest empirical mean in non-exploration steps. Thus, we only need to consider the exploration steps, i.e., steps during which we choose to explore arms according to round-robin. Let $t_i^\varepsilon(k)$ be the time slot that we explore arm $i$ for the $k$-th time. Then the compensation the controller has to pay in this time slot is $\mathbb{E}[\max_j \hat{\mu}_j(t_i^\varepsilon(k)) - \hat{\mu}_i(t_i^\varepsilon(k))]$.

Since the rewards are independent of whether we choose to explore, one sees that $\mathbb{E}[\hat{\mu}_i(t_i^\varepsilon(k))] = \mu_i$. Thus, we can decompose $\mathbb{E}[\max_j \hat{\mu}_j(t_i^\varepsilon(k)) - \hat{\mu}_i(t_i^\varepsilon(k))]$ as follows:

$$\begin{aligned} \mathbb{E}[\max_j \hat{\mu}_j(t_i^\varepsilon(k)) - \hat{\mu}_i(t_i^\varepsilon(k))] &= \mathbb{E}[\max_j(\hat{\mu}_j(t_i^\varepsilon(k)) - \mu_i)] \\ &\leq \mathbb{E}[\max_j(\hat{\mu}_j(t_i^\varepsilon(k)) - \mu_j)] + \mathbb{E}[\max_j(\mu_j - \mu_i)]. \end{aligned} \tag{2}$$

The second term in (2) is bounded by $\Delta_i = \mu_1 - \mu_i$. Summing over all these steps and all arms, we obtain the first term $\sum_{i=2}^{N} \frac{c\Delta_i \log T}{\Delta_2^2}$ in our bound (1).

We turn to the first term in (2), i.e., $\mathbb{E}[\max_j(\hat{\mu}_j(t_i^\varepsilon(k)) - \mu_j)]$. We see that it is upper bounded by

$$\mathbb{E}[\max_j(\hat{\mu}_j(t_i^\varepsilon(k)) - \mu_j)] \leq \mathbb{E}[\max_j(\hat{\mu}_j(t_i^\varepsilon(k)) - \mu_j)^+] \leq \sum_j \mathbb{E}[(\hat{\mu}_j(t_i^\varepsilon(k)) - \mu_j)^+]$$

where $(*)^+ = \max\{*, 0\}$. When arm $i$ has been explored $k$ times (line 7 in Algorithm 2), we know that all other arms have at least $k$ observations (in the first $N$ time slots, there is one observation for each arm). Hence, $\mathbb{E}[(\hat{\mu}_j(t_i^\varepsilon(k)) - \mu_j)^+] = \frac{1}{2}\mathbb{E}[|\hat{\mu}_j(t_i^\varepsilon(k)) - \mu_j|] \leq \frac{1}{4\sqrt{k}}$ (the equality is due to the fact that $\mathbb{E}[|x|] = 2\mathbb{E}[x^+]$ if $\mathbb{E}[x] = 0$).

Suppose arm $i$ is been explored in time set $T_i = \{t_i^\varepsilon(1), \cdots\}$. Then,

$$\sum_{k \leq |T_i|} \mathbb{E}[\max_j(\hat{\mu}_j(t_i^\varepsilon(k)) - \mu_j)^+] \leq \sum_{k \leq |T_i|} \frac{N}{4\sqrt{k}} \leq \frac{N\sqrt{|T_i|}}{2}$$

Since $\mathbb{E}[|T_i|] = \frac{c}{\Delta_2^2} \log T$, we can bound the first term in (2) as $\frac{N^2\sqrt{c \log T}}{2\Delta_2}$. Summing this with $\sum_{i=2}^{N} \frac{c\Delta_i \log T}{\Delta_2^2}$ above for the second term, we obtain the compensation upper bound in (1). $\square$

```
 1: Init: $\alpha_i = 1, \beta_i = 1$ for each arm $i$.
 2: for $t = 1, 2, \cdots, N$ do
 3:     Choose arm $a(t) = t$ and receive the observation $X(t)$.
 4:     Update$(\alpha_{a(t)}, \beta_{a(t)}, X(t))$
 5: end for
 6: for $t = N + 1, N + 3, \cdots$ do
 7:     For all $i$ sample values $\theta_i(t)$ from Beta distribution $\mathbf{B}(\alpha_i, \beta_i)$;
 8:     Play action $a_1(t) = \operatorname{argmax}_i \hat{\mu}_i(t)$, get the observation $X(t)$. Update$(\alpha_{a_1(t)}, \beta_{a_1(t)}, X(t))$
 9:     Play action $a_2(t+1) = \operatorname{argmax}_i \theta_i(t)$ (with compensation $\max_j \hat{\mu}_j(t+1) - \hat{\mu}_{a_2(t+1)}(t+1)$),
        receive the observation $X(t + 1)$. Update$(\alpha_{a_2(t+1)}, \beta_{a_2(t+1)}, X(t + 1))$
10: end for
```

**Algorithm 3:** The Modified Thompson Sampling Algorithm for KCMAB.

```
 1: Input: $\alpha_i, \beta_i, X(t)$
 2: Output: updated $\alpha_i, \beta_i$
 3: $Y(t) \leftarrow 1$ with probability $X(t)$, 0 with probability $1 - X(t)$
 4: $\alpha_i \leftarrow \alpha_i + Y(t); \beta_i \leftarrow \beta_i + 1 - Y(t)$
```

**Algorithm 4:** Procedure Update

## 4.3 The Modified Thompson Sampling policy

The third algorithm we propose is a Thompson Sampling (TS) based policy. Due to the complexity of the analysis for the traditional TS algorithm, we propose a modified TS policy and derive its compensation bound. Our modification is motivated by the LUCB algorithm [10]. Specifically, we divide time into rounds containing two time steps each, and pull not only the arm with largest sample value, but also the arm with largest empirical mean in each round. The modified TS policy is presented in Algorithm 3, and we have the following theorem about its regret and compensation.

**Theorem 4.** *In Algorithm 3, we have*

$$Reg(T) \le \sum_i \frac{2\Delta_i}{(\Delta_i - \varepsilon)^2} \log T + O\left(\frac{N}{\varepsilon^4}\right) + F_1(\boldsymbol{\mu})$$

*for some small $\varepsilon < \Delta_2$ and $F_1(\boldsymbol{\mu})$ does not depend on $(T, \varepsilon)$. As for compensation, we have:*

$$Com(T) \le \sum_i \frac{8}{\Delta_i - \varepsilon} \log T + N \log T + O\left(\frac{N}{\varepsilon^4}\right) + F_2(\boldsymbol{\mu})$$

*where $F_2(\boldsymbol{\mu})$ does not depend on $(T, \varepsilon)$ as well.*

**Proof Sketch:** In round $(t, t + 1)$, we assume that we first run the arm with largest empirical mean on time slot $t$ and call $t$ an empirical step. Then we run the arm with largest sample on time slot $t + 1$ and call $t + 1$ a sample step.

We can bound the number of sample steps during which we pull a sub-optimal arm, using existing results in [1], since all sample steps form an approximation of the classic TS algorithm. Moreover, [11] shows that in sample steps, the optimal arm is pulled for many times (at least $t^b$ at time $t$ with a constant $b \in (0, 1)$). Thus, after several steps, the empirical mean of the optimal arm will be accurate enough. Then, if we choose to pull sub-optimal arm $i$ during empirical steps, arm $i$ must have an inaccurate empirical mean. Since the pulling will update its empirical mean, it is harder and harder for the arm's empirical mean to remain inaccurate. As a result, it cannot be pulled a lot of times during the empirical steps as well.

Next, we discuss how to bound its compensation. It can be shown that with high probability, we always have $|\theta_i(t) - \hat{\mu}_i(t)| \le r_i(t)$, where $r_i(t) = \sqrt{\frac{2 \log t}{N_i(t)}}$ is defined in Algorithm 1. Thus, we can focus on the case that $|\theta_i(t) - \hat{\mu}_i(t)| \le r_i(t)$ for any $i$ and $t$. Note that we do not need to pay compensation in empirical steps. In sample steps, suppose we pull arm $i$ and the largest empirical mean is in arm $j \ne i$ at the beginning of this round. Then, we need to pay $\max_k \hat{\mu}_k(t+1) - \hat{\mu}_i(t+1)$, which is upper bounded by $\hat{\mu}_j(t) - \hat{\mu}_i(t) + (\hat{\mu}_j(t + 1) - \hat{\mu}_j(t))^+ \le \hat{\mu}_j(t) - \hat{\mu}_i(t) + \frac{1}{N_j(t)}$ (here

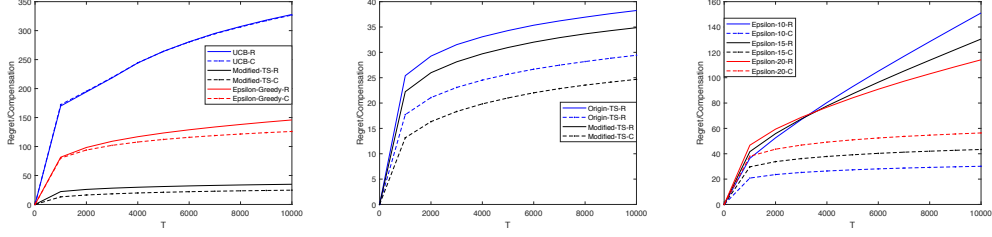

Figure 1: Regret and Compensa-tion of Three policies.
Figure 2: Regret and Compensa-tion of TS and modified-TS.
Figure 3: Regret and Compensa-tion of modified $\varepsilon$-greedy.

$\hat{\mu}_i(t+1) = \hat{\mu}_i(t)$). As $\theta_i(t) \geq \theta_j(t)$, we must have $\hat{\mu}_i(t) + r_i(t) \geq \theta_i(t) \geq \theta_j(t) \geq \hat{\mu}_j(t) - r_j(t)$, which implies $\hat{\mu}_j(t) - \hat{\mu}_i(t) \leq r_i(t) + r_j(t)$. Thus, what we need to pay is at most $r_i(t) + r_j(t) + \frac{1}{N_j(t)}$ if $i \neq j$, in which case we can safely assume that we pay $r_j(t) + \frac{1}{N_j(t)}$ during empirical steps, and $r_i(t)$ during sample steps.

For an sub-optimal arm $i$, we have $Com_i(T) \leq \sum_i \frac{4}{\Delta_i - \varepsilon} \log T + O(\frac{1}{\varepsilon^4}) + F_1(\boldsymbol{\mu}) + \log T$ (summing over $r_i(t)$ gives the same result as in the UCB case, and summing over $\frac{1}{N_i(t)}$ is upper bounded by $\log T$). As for arm 1, when $a_1(t) = a_2(t+1) = 1$, we do not need to pay $r_1(t)$ twice. In fact, we only need to pay at most $\frac{1}{N_1(t)}$. Then, the number of time steps that $a_1(t) = a_2(t+1) = 1$ does not happen is upper bounded by $\sum_{i=2}^{N} \left( \frac{2}{(\Delta_i - \varepsilon)^2} \log T \right) + O\left( \frac{N}{\varepsilon^4} \right) + F_1(\boldsymbol{\mu})$, which is given by regret analysis. Thus, the compensation we need to pay on arm 1 is upper bounded by $\sum_i \frac{4}{\Delta_i - \varepsilon} \log T + O(\frac{1}{\varepsilon^4}) + F_1(\boldsymbol{\mu}) + \log T$. Combining the above, we have the compensation bound $Com(T) \leq \sum_i \frac{8}{\Delta_i - \varepsilon} \log T + N \log T + O(\frac{1}{\varepsilon^4}) + F_2(\boldsymbol{\mu})$. $\square$

# 5   Experiments

In this section, we present experimental results for the three algorithms, i.e., the UCB policy, the modified $\varepsilon$-greedy policy and the modified TS policy. We also compare our modified TS policy with origin TS policy to evaluate their difference. In our experiments, there are a total of nine arms with expected reward vector $\boldsymbol{\mu} = [0.9, 0.8, 0.7, 0.6, 0.5, 0.4, 0.3, 0.2, 0.1]$. We run the game for $T = 10000$ time steps. The experiment runs for 1000 times and we take the average over these results. The "-R" represents the regret of that policy, and "-C" represents the compensation.

The comparison of the three policies in this paper is shown in Figure 1. We can see that modified-TS performs best in both regret and compensation, compared to other algorithms. As for the modified $\varepsilon$-greedy policy, when the parameter $\epsilon$ is chosen properly, it can also achieve a good performance. In our experiment, we choose $\epsilon = 20$.

In Figure 2, we see that modified-TS performs better than TS in both compensation and regret, which means that our modification is effective. Figure 3 shows the different performance of the modified $\varepsilon$-greedy policies with different $\epsilon$ values. Here we choose $\epsilon$ to be 10,15 and 20. From the experiments, we see the trade-off between regret and compensation: low compensation leads to high regret, and high compensation leads to low regret.

# 6   Conclusion

We propose and study the known-compensation multi-armed bandit (KCMAB) problem where a controller offers compensation to incentivize players for arm exploration. We first establish a compensation lower bound achieved by regret-minimizing algorithms. Then, we consider three algorithms, namely, UCB, modified $\varepsilon$-greedy and modified TS. We show that all three algorithms achieve good regret bounds, while keeping order-optimal compensation. We also conduct experiments and the results validate our theoretical findings.

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
