[Supplementary Material · nips_2019_01_10_supplementary.pdf]

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

## Supplementary Material

The proofs of all lemmas are shown in the end of the sections they belong to.

## A    Proof of Theorem 1

**Fact 2.** *(Theorem 2 in [12]) If an algorithm guarantees $o(T^{\alpha})$ regret upper bound for any $\alpha > 0$, then for any $\varepsilon > 0$, we have:*

$$\lim_{T \to \infty} \Pr[N_i(t) \geq (1 - \varepsilon) \frac{\log T}{KL(D_i, D_1)}] = 1$$

Fact 2 means that we will pull arm $i$ for at least $(1 - \varepsilon) \frac{\log T}{KL(D_i, D_1)}$ times almost surely, i.e. there exists a function $\delta(T) \to 0$ as $T \to \infty$ such that $\Pr[N_i(T) \geq (1 - \varepsilon) \frac{\log T}{KL(D_i, D_1)}] \geq 1 - \delta(T)$.

Consider the time step when we pull arm $i$ for the $k$-th time ($2 \leq k \leq (1 - \varepsilon) \frac{\log T}{KL(D_i, D_1)}$), and we use $t_i(k)$ to denote the random variable of this time slot. Since we may not pull arm $i$ for $k$ times until time $T$, we suppose that the game lasts for infinite number of times, and $t_i(k)$ can be larger than $T$. By the definition of $t_i(k)$, we must have $\mathbb{E}[\hat{\mu}_i(t_i(k))] = \mu_i$.

Then the compensation we need to pay for pulling arm $i$ for the $k$-th time can be bounded as following:

$$
\begin{aligned}
\Pr[t_i(k) \leq T]\mathbb{E}[c(t_i(k))|t_i(k) \leq T] &= \Pr[t_i(k) \leq T]\mathbb{E}[\max_j \hat{\mu}_j(t_i(k)) - \hat{\mu}_i(t_i(k))|t_i(k) \leq T] \\
&\geq \mathbb{E}[\max_j \hat{\mu}_j(t_i(k)) - \hat{\mu}_i(t_i(k))] - \Pr[t_i(k) > T] \quad (3) \\
&\geq \mathbb{E}[\max_j \hat{\mu}_j(t_i(k))] - \mathbb{E}[\hat{\mu}_i(t_i(k))] - \delta(T) \quad (4) \\
&\geq \mathbb{E}[\hat{\mu}_1(t_i(k))] - \mathbb{E}[\hat{\mu}_i(t_i(k))] - \delta(T) \\
&= \mathbb{E}[\hat{\mu}_1(t_i(k))] - \mu_i - \delta(T) \quad (5)
\end{aligned}
$$

Eq. (3) is because that $\max_j \hat{\mu}_j(t_i(k)) - \hat{\mu}_i(t_i(k)) \leq 1$, and Eq. (4) is because that $\Pr[t_i(k) > T] \leq \delta(T)$, which is given by Fact 2.

In Theorem 1, we suppose that $D_1$ is a Bernoulli distribution, i.e. $\Pr[X_1(t) = 1] = \mu_1$ and $\Pr[X_1(t) = 0] = 1 - \mu_1$. Then we can use a 0-1 string $s$ to represent the history of arm 1. In this case $\hat{\mu}_1(t_i(k)) = \frac{\#(s)}{|s|}$, where $\#(s)$ is the number of 1s in string $s$. To simplify the notations, we use $z - s$ to denote the string that removes prefix $s$ from $z$, and $s + z$ to denote the string given by adding prefix $s$ to $z$.

We can thus rewrite $\mathbb{E}[\hat{\mu}_1(t_i(k))]$ as:

$$\mathbb{E}[\hat{\mu}_1(t_i(k))] = \sum_s p(s) \frac{\#(s)}{|s|},$$

where $p(s)$ is the probability that at time slot $t_i(k)$, the history of arm 1 forms string $s$. Note that this expectation is hard to evaluate, since all the arms are coupled due to the algorithms, which makes $\hat{\mu}_1(t_i(k))$ dependent on $t_i(k)$.

We define two events as following: $\mathcal{A}_L(s)$ is the event that the first $|s|$ feedbacks of arm 1 form string $s$, and $\mathcal{B}_L(s)$ is the event that at time slot $t_i(k)$, the feedbacks of arm 1 form string $s$. Then $\Pr[\mathcal{A}_L(s)] = P(s, \mu_1)$, where $P(s, \mu_1) = \mu_1^{\#(s)}(1 - \mu_1)^{|s| - \#(s)}$, and $\Pr[\mathcal{B}_L(s)] = p(s)$.

In our model, we suppose that we pull each arm once in the first $N$ time slots, then $\sum_{|s| = 0} p(s) = 0$, which implies $\sum_{|s| > 0} p(s) = 1$, thus $p$ is a probability distribution on all 0-1 strings. Since $p$ is a probability distribution, when the first $|s|$ feedbacks of arm 1 form string $s$, there must be a string $z$ such that $\mathcal{B}_L(z)$ happens and either $z$ is prefix of $s$ or $s$ is prefix of $z$. Moreover, if $z$ is prefix of $s$, we still need the next $|s| - |z|$ feedbacks from arm 1 form the string $s - z$. This means the following equation holds:

$$\mathcal{A}_L(s) = \cup_{z: pre(z, |s|) = s} \mathcal{B}_L(z) \cup_{y \in sub(s)} (\mathcal{B}_L(y) \cap \{\text{The next } |s| - |y| \text{ feedbacks form string } s - y\}),$$

where $pre(z, n)$ is the prefix of $z$ with length $n$, and $sub(s) = \{pre(s, j) | 1 \le j \le |s| - 1\}$ is the set that contains all prefixes of $s$ but does not contain $s$ itself.

For any $s \ne z$, we must have $\mathcal{B}_L(s) \cap \mathcal{B}_L(z) = \emptyset$, thus

$$
\begin{aligned}
\Pr[\mathcal{A}_L(s)] &= \sum_{z:pre(z,|s|)=s} \Pr[\mathcal{B}_L(z)] + \sum_{y \in sub(s)} \Pr[\mathcal{B}_L(y) \cap \{\text{The next } |s| - |y| \text{ feedbacks forms string } s - y\}] \\
&= \sum_{z:pre(z,|s|)=s} p(z) + \sum_{y \in sub(s)} p(y)P(s - y, \mu_1)
\end{aligned}
\tag{6}
$$

Eq. (6) is because that the next $|s| - |y|$ feedbacks are independent with event $\mathcal{B}_L(y)$. From this equation, we have $P(s, \mu_1) = \sum_{z:pre(z,|s|)=s} p(z) + \sum_{y \in sub(s)} p(y)P(s - y, \mu_1)$.

To simplify the analysis, we construct $p^T$ from $p$ by adding the probability of $p(s)$ with $s > T$ to $p(pre(s, T))$:

$$
p^T(s) = \begin{cases} p(s) & 1 \le |s| < T \\ p(s) + \sum_{z:pre(z,T)=s} p(z) & |s| = T \\ 0 & |s| > T \end{cases}
$$

This does not influence the Eq. (6) for all $s$ with $|s| \le T$ (the probability mass for strings longer than $T$ is included in its $T$-sized prefix). Thus $P(s, \mu_1) = \sum_{z:pre(z,|s|)=s} p^T(z) + \sum_{y \in sub(s)} p^T(y)P(s - y, \mu_1)$ still holds.

Now we can use $p^T$ to bound $\mathbb{E}[\hat{\mu}_1(t_i(k))]$ as:

$$
\mathbb{E}[\hat{\mu}_1(t_i(k))] = \sum_s p(s)\frac{\#(s)}{|s|} \ge \sum_s p^T(s)\frac{\#(s)}{|s|} - \delta(T)
\tag{7}
$$

Here $\sum_{|s|>T} p(s) \le \delta(T)$ holds by Fact 2 ($\cup_{|s|>T}\mathcal{B}_L(s)$ implies $N_i(T) \le k \le (1 - \varepsilon)\frac{\log T}{KL(D_i, D_1)}$).

From $p^T$, we can then build another $q^T$ as follows.

$$
q^T(s) = \begin{cases} \frac{p^T(s)}{P(s,\mu_1) - \sum_{y \in sub(s)} p^T(y)P(s-y,\mu_1)} & \text{if } P(s, \mu_1) - \sum_{y \in sub(s)} p^T(y)P(s - y, \mu_1) > 0 \\ 1 & \text{if } P(s, \mu_1) - \sum_{y \in sub(s)} p^T(y)P(s - y, \mu_1) = 0 \end{cases}
$$

One can check that $q^T(s) = 1$ for any $|s| = T$ and $0 \le q^T(s) \le 1$ for any $1 \le |s| < T$.

**Lemma 1.** *We can get the value of $p^T$ from $q^T$ by the following equation,*

$$
p^T(s) = P(s, \mu_1) \prod_{j=1}^{|s|-1} (1 - q^T(pre(s, j)))q^T(s)
\tag{8}
$$

Lemma 1 means that every possible $p^T$ has a unique $q^T$ match and vice versa. Then we can consider a set of $q^T(s)$ such that the corresponding $p^T(s)$ minimizes $\sum_{1 \le |s| \le T} p^T(s)\frac{\#(s)}{|s|}$. We write $emp(q^T, \mu_1, T)$ to denote the corresponding value $\sum_{1 \le |s| \le T} p^T(s)\frac{\#(s)}{|s|}$. Then for given $q^T$, we can use Algorithm 5 to compute $emp(q^T, \mu_1, T)$.

**Proposition 1.** *Algorithm 5 returns the value $emp(q^T, \mu_1, T)$ correctly.*

To find $q^T$ that minimizes $emp(q^T, \mu_1, T)$, we introduce a Dynamic Programming policy as Algorithm 6. It starts by setting $f(s) = \frac{\#(s)}{|s|}$ for all strings with $|s| = T$. After that, if all strings $s$ with $|s| = k$ have their $f(s)$, we start to consider $s$ with $|s| = k - 1$. For any $|s| = k - 1$, the DP policy will check whether it is good or not to stop at $s$, i.e. only if $\frac{\#(s)}{|s|}$ is smaller than

```
1:  Input:  q^T, μ_1, T.
2:  for s with |s| = T do
3:      g(s) ← #(s)/|s|.
4:  end for
5:  for t = T − 1, ⋯ , 1 do
6:      for s with |s| = t do
7:          g(s) ← q^T(s) #(s)/|s| + (1 − q^T(s))(μg(s + "1") + (1 − μ)g(s + "0"))
8:      end for
9:  end for
10: emp(q^T, μ_1, T) = μg("1") + (1 − μ)g("0")
11: Output:  emp(q^T, μ_1, T).
```

**Algorithm 5:** Calculate $emp(q^T, \mu_1, T)$

```
1:  Input:  μ_1, T.
2:  for a = 0, 1, ⋯ , T do
3:      f(a, T − a) ← a/T.
4:  end for
5:  for t = T − 1, ⋯ , 1 do
6:      for a = 0, 1, ⋯ , t do
7:          f(a, t − a) ← min{a/t, μf(a + 1, t − a) + (1 − μ)f(a, t − a + 1)}
8:      end for
9:  end for
10: f(0, 0) = μf(1, 0) + (1 − μ)f(0, 1)
11: Output:  DP(μ_1, T) = f(0, 0)
```

**Algorithm 6:** Dynamic Programming stopping policy

$\mu_1 f(s + "1") + (1 − \mu_1)f(s + "0")$, we choose to set $q^T(s) = 1$ and $f(s) = \frac{\#(s)}{|s|}$, otherwise we set $q^T(s) = 0$ and $f(s) = \mu_1 f(s + "1") + (1 − \mu_1)f(s + "0")$.

Intuitively, the DP policy is the best one can do, which is shown in the following lemma.

**Lemma 2.** *For any given $q^T$, $emp(q^T, \mu_1, T) \geq DP(\mu_1, T)$, where $DP(\mu_1, T)$ is the output value of Algorithm 6.*

Now from Eq. (7) and Lemma 2, we have:

$$\mathbb{E}[\hat{\mu}_1(t_i(k))] \geq DP(\mu_1, T) − \delta(T) \tag{9}$$

Then we need a lower bound on $DP(\mu_1, T)$, which is given in the following lemma.

**Lemma 3.** *For any $\mu_1 \geq 0.9$ and $T \geq 1$, we have $DP(\mu_1, T) \geq \frac{\mu_1}{2}$*

From Lemma 3, Eq. (9) and Eq. (5), we know that when $\mu_1 \geq 0.9$, $0.2 \geq \mu_2 \geq \mu_3 \geq \cdots \geq \mu_N$, $\mathbb{E}[c(t_i(k))] \geq DP(\mu_1, T) − \mu_i − 2\delta(T) \geq \frac{\Delta_i}{4} − 2\delta(T)$ for any $i$ and $2 \leq k \leq (1 − \varepsilon)\frac{\log T}{KL(D_i, D_1)}$. Thus the compensation we need to pay is $\Omega(\sum_{i=2}^{N}(\Delta_i − 2\delta(T))\frac{\log T}{KL(D_i, D_1)})$. When $T \to \infty$, we have $\delta(T) \to 0$. Therefore, the compensation is lower bounded by $\Omega(\sum_{i=2}^{N}\frac{\Delta_i \log T}{KL(D_i, D_1)})$.

### A.1  Proof of Lemma 1

We prove this lemma by induction.

For the strings $s$ with $|s| = 1$, since there is no string in $sub(s)$, we have $q^T(s) = \frac{p^T(s)}{P(s, \mu_1)}$ by definition of $q^T$. Thus Eq. (8) holds.

If for all strings $s$ with $|s| \leq k$, Eq. (8) holds. Then consider a string $s$ with $|s| = k + 1$, we choose $z$ as the longest string in $sub(s)$ such that $q^T(z) > 0$.

If such $z$ does not exist. Then by definition of $q^T(z)$, we know $p^T(z) = 0$ for all $z \in sub(s)$. Thus we have $P(s, \mu_1) \prod_{j=1}^{|s|-1} (1 - q^T(pre(s, j))) q^T(s) = P(s, \mu_1) q^T(s) = p^T(s)$ holds.

When such $z$ exists, let $x = s - z$, and then we have

$$P(s, \mu_1) \prod_{j=1}^{|s|-1} (1 - q^T(pre(s, j))) q^T(s)$$

$$= \left( P(z, \mu_1) \prod_{j=1}^{|z|-1} (1 - q^T(pre(z, j))) \right) (1 - q^T(z)) q^T(s) P(x, \mu_1)$$

$$= \frac{p^T(z)}{q^T(z)} (1 - q^T(z)) q^T(s) P(x, \mu_1) \tag{10}$$

Eq. (10) is because that by induction, Eq. (8) holds for any $|z| \le k$.

If $P(z, \mu_1) - \sum_{y \in sub(z)} p^T(y) P(z - y, \mu_1) = 0$, then we must have $p^T(z) = 0$, thus $P(s, \mu_1) - \sum_{y \in sub(s)} p^T(y) P(s - y, \mu_1) \le P(x, \mu_1) \left( P(z, \mu_1) - \sum_{y \in sub(z)} p^T(y) P(z - y, \mu_1) \right) = 0$, which means $p^T(s) = 0$. On the other hand, $P(z, \mu_1) - \sum_{y \in sub(z)} p^T(y) P(z - y, \mu_1) = 0$ means $q^T(z) = 1$, thus Eq. (8) holds.

If $P(z, \mu_1) - \sum_{y \in sub(z)} p^T(y) P(z - y, \mu_1) > 0$, then

$$\frac{p^T(z)}{q^T(z)} (1 - q^T(z)) q^T(s) P(x, \mu_1)$$

$$= p^T(z) \frac{1 - q^T(z)}{q^T(z)} q^T(s) P(x, \mu_1)$$

$$= p^T(z) \left( \frac{P(z, \mu_1) - \sum_{y \in sub(z)} p^T(y) P(z - y, \mu_1)}{p^T(z)} - 1 \right) q^T(s) P(x, \mu_1)$$

$$= \left( P(z, \mu_1) - \sum_{y \in sub(z)} p^T(y) P(z - y, \mu_1) - p^T(z) \right) q^T(s) P(x, \mu_1)$$

$$= \left( P(s, \mu_1) - \sum_{y \in sub(s)} p^T(y) P(s - y, \mu_1) \right) q^T(s) \tag{11}$$

$$= p^T(s)$$

Eq. (11) is because that for all $s' \in sub(s)$ and $|s'| > |z|$, $q^T(s') = 0$ implies $p^T(s') = 0$.

By induction, we finish the proof of Lemma 1.

### A.2 Proof of Lemma 2

We use induction to prove that $emp(q^T, \mu_1, T) \ge DP(\mu_1, T)$.

For all $s$ with size $|s| = T$, we can see that $f(\#(s), |s| - \#(s)) = \frac{\#(s)}{|s|} \le g(s)$, where $f(\#(s), |s| - \#(s))$ is the value in Algorithm 6 with input $\mu_1, T$.

If for all $s$ with size $|s| = k$, we have $f(\#(s), |s| - \#(s)) \le g(s)$, then consider any $s'$ with size $|s'| = k - 1$.

$$f(\#(s'), |s'| - \#(s')) \quad = \quad \min\{\frac{\#(s')}{|s'|}, \mu f(\#(s) + 1, |s| - \#(s)) + (1 - \mu) f(\#(s), |s| - \#(s) + 1)\}$$

$$\leq \quad \min\{\frac{\#(s')}{|s'|}, \mu g(s' + \text{"1"}) + (1 - \mu)g(s' + \text{"0"})\}$$

$$\leq \quad g(s')$$

Thus by induction, we have $f(0,1) \leq g(\text{"0"})$ and $f(1,0) \leq g(\text{"1"})$, which means $emp(q^T, \mu_1, T) \geq DP(\mu_1, T)$.

### A.3 Proof of Lemma 3

We calculate this by summing over the difference between $DP(\mu_1, T)$ and $DP(\mu_1, T+1)$. Let $f^{\mu_1,T}(a,b) - f^{\mu_1,T+1}(a,b) = \delta_{a,b}^{\mu_1,T}$, where $f^{\mu,T}(a,b)$ is the value of $f(a,b)$ when inputting $\mu, T$ into Algorithm 6.

First consider the case that $a + b = T$, $f^{\mu_1,T}(a,b) = \frac{a}{T}$, while $f^{\mu_1,T+1}(a,b) = \min\{\frac{a}{T}, \mu_1\frac{a+1}{T+1} + (1-\mu_1)\frac{a}{T+1}\}$.

Since $\mu_1 \frac{a+1}{T+1} + (1-\mu_1)\frac{a}{T+1} = \frac{a+\mu_1}{T+1} = \frac{a}{T} + \frac{\mu_1 - a/T}{T+1} = \frac{a}{T} + \frac{\mu_1 T - a}{T(T+1)}$. When $a > \mu_1 T$, we have $\delta_{a,b}^{\mu_1,T} = \frac{a-\mu_1 T}{T(T+1)}$; otherwise $\delta_{a,b}^{\mu_1,T} = 0$. Thus $\delta_{a,b}^{\mu_1,T} = \frac{1}{T(T+1)}(a - \mu_1 T)^+$.

Then we consider the case that $a + b = t < T$. By definition, $f^{\mu_1,T}(a,b) = \min\{\frac{a}{t}, \mu_1 f^{\mu_1,T}(a+1,b) + (1-\mu_1)f^{\mu_1,T}(a,b+1)\}$. Thus

$$
\begin{aligned}
\delta_{a,b}^{\mu_1,T} &= f^{\mu_1,T}(a,b) - f^{\mu_1,T+1}(a,b) \\
&= \min\{\frac{a}{t}, \mu_1 f^{\mu_1,T}(a+1,b) + (1-\mu_1)f^{\mu_1,T}(a,b+1)\} \\
&\quad - \min\{\frac{a}{t}, \mu_1 f^{\mu_1,T+1}(a+1,b) + (1-\mu_1)f^{\mu_1,T+1}(a,b+1)\} \\
&\leq (\mu_1 f^{\mu_1,T}(a+1,b) + (1-\mu_1)f^{\mu_1,T}(a,b+1)) \\
&\quad -(\mu_1 f^{\mu_1,T+1}(a+1,b) + (1-\mu_1)f^{\mu_1,T+1}(a,b+1)) \\
&= \mu_1 \delta_{a+1,b}^{\mu_1,T} + (1-\mu_1)\delta_{a,b+1}^{\mu_1,T}
\end{aligned}
\tag{12}
$$

Eq. (12) is because of the fact that $f^{\mu_1,T}(a,b) \geq f^{\mu_1,T+1}(a,b)$ for any given $(a,b)$.

This implies :

$$
\begin{aligned}
\delta_{0,0}^{\mu_1,T} &\leq \mu_1 \delta_{1,0}^{\mu_1,T} + (1-\mu_1)\delta_{0,1}^{\mu_1,T} \\
&\leq \mu_1^2 \delta_{2,0}^{\mu_1,T} + 2\mu_1(1-\mu_1)\delta_{1,1}^{\mu_1,T} + (1-\mu_1)^2 \delta_{0,2}^{\mu_1,T} \\
&\leq \sum_{a+b=3} \binom{3}{a} \mu_1^a (1-\mu_1)^b \delta_{a,b}^{\mu_1,T} \\
&\leq \cdots \\
&\leq \sum_{a+b=T} \binom{T}{a} \mu_1^a (1-\mu_1)^b \delta_{a,b}^{\mu_1,T} \\
&= \mathbb{E}[\frac{1}{T(T+1)}(a - \mu_1 T)^+] \\
&= \frac{1}{T(T+1)}\mathbb{E}[(a - \mu_1 T)^+]
\end{aligned}
$$

The expectation is taken over a binomial distribution $a \sim Binomial(T, \mu_1)$.

Since $\mathbb{E}[a] = \mu_1 T$, we have that:

$$\mathbb{E}[(a - \mu_1 T)^+] = \frac{1}{2}\mathbb{E}[|a - \mu_1 T|] \leq \frac{1}{2}\sqrt{\mathbb{E}[(a - \mu_1 T)^2]} = \frac{1}{2}\sqrt{T\mu_1(1-\mu_1)},$$

which leads to the upper bound $\delta_{0,0}^{\mu_1,T} \leq \frac{\sqrt{\mu_1(1-\mu_1)}}{2(T+1)\sqrt{T}}$.

Thus,

$$
\begin{aligned}
DP(\mu_1, T) &\geq DP(\mu_1, 1) - \sum_{T=1}^{\infty} \frac{\sqrt{\mu_1(1-\mu_1)}}{2(T+1)\sqrt{T}} \\
&\geq \mu_1 - \frac{\sqrt{\mu_1(1-\mu_1)}}{2} \sum_{T=1}^{\infty} \frac{1}{T^{3/2}} \\
&\geq \mu_1 - \frac{\sqrt{\mu_1(1-\mu_1)}}{2}(1 + \int_1^{\infty} \frac{1}{T^{3/2}} dT) \\
&= \mu_1 - \frac{\sqrt{\mu_1(1-\mu_1)}}{2}(1+2) \\
&= \mu_1 - \frac{3\sqrt{\mu_1(1-\mu_1)}}{2}
\end{aligned}
$$

When $\mu_1 \geq 0.9$, $\frac{3\sqrt{\mu_1(1-\mu_1)}}{2} \leq 0.45$, thus $DP(\mu_1, T) \geq \mu_1 - 0.45 \geq \frac{\mu_1}{2}$.

## B  Proof of Theorem 1 when $T$ is unknown

This proof is suggested by an anonymous reviewer of our paper during the review process. We thank the reviewer for the idea.

**Proposition 2.** *In KCMAB, if an algorithm guarantees an $o(T^\alpha)$ regret upper bound for any $T$ and any $\alpha > 0$, then the algorithm must pay $\Omega\left(\sum_{i=2}^{N} \frac{\Delta_i \log T}{KL(D_i, D_1)}\right)$ for compensation.*

Choose $N_1^*(\varepsilon) = \frac{9}{2\Delta_2^2} \log \frac{9}{\varepsilon \Delta_2^2}$, and $N_1^{**}(\varepsilon)$ be the time step such that with probability $1 - \frac{\varepsilon}{2}$, $N_1(t) > \frac{t}{2}$ for any $t \geq N_1^{**}(\varepsilon)$. Notice that $N_1^{**}(\varepsilon)$ must exists since the algorithm has $o(T)$ regret in expectation, and it does not depend on $T$.

Now choose $T_1^*(\varepsilon) = \max\{2N_1^*(\varepsilon), N_1^{**}(\varepsilon)\}$. Note that $T_1^*(\varepsilon)$ does not depend on $T$ as well. The probability that $\{\exists t > T_1^*(\varepsilon), \hat{\mu}_1(t) < \mu_1 - \frac{\Delta_2}{3}\}$ happens can be upper bounded by:

$$
\begin{aligned}
&\Pr[\exists t > T_1^*(\varepsilon), \hat{\mu}_1(t) < \mu_1 - \frac{\Delta_2}{3}] \\
\leq\ & \Pr[N_1(T_1^*(\varepsilon)) \leq N_1^*(\varepsilon)] + \Pr[\{N_1(T_1^*(\varepsilon)) > N_1^*(\varepsilon)\} \wedge \{\exists t > T_1^*(\varepsilon), \hat{\mu}_1(t) < \mu_1 - \frac{\Delta_2}{3}\}] \\
\leq\ & \Pr[N_1(T_1^*(\varepsilon)) \leq N_1^*(\varepsilon)] + \Pr[\exists n > N_1^*(\varepsilon) \text{ s.t. } N_1(t) = n, \hat{\mu}_1(t) < \mu_1 - \frac{\Delta_2}{3}] \\
\leq\ & \Pr[N_1(T_1^*(\varepsilon)) \leq N_1^*(\varepsilon)] + \sum_{n > N_1^*(\varepsilon)} \left( \Pr[N_1(t) = n, \hat{\mu}_1(t) < \mu_1 - \frac{\Delta_2}{3}] \right) \quad (13)
\end{aligned}
$$

The first term in (13) has upper bound $\frac{\varepsilon}{2}$ by definition of $N_1^{**}(\varepsilon)$ and $T_1^*(\varepsilon)$.

As for the second term in (13), notice that $\{N_1(t) = n, \hat{\mu}_1(t) < \mu_1 - \frac{\Delta_2}{3}\}$ implies the first $n$ feedbacks of arm 1 have an empirical mean less than $\mu_1 - \frac{\Delta_2}{3}$. By Chernoff Bound, $\Pr[N_1(t) = n, \hat{\mu}_1(t) < \mu_1 - \frac{\Delta_2}{3}] \leq \exp(-2n\Delta_2^2/9)$. Therefore $\sum_{n > N_1^*(\varepsilon)} \Pr[N_1(t) = n, \hat{\mu}_1(t) < \mu_1 - \frac{\Delta_2}{3}] \leq \frac{9}{2\Delta_2^2} \exp(-2N_1^*(\varepsilon)\Delta_2^2/9) \leq \frac{\varepsilon}{2}$.

This means that with probability at least $1 - \varepsilon$, for any $t > T_1^*(\varepsilon)$, $\hat{\mu}_1(t) \geq \mu_1 - \frac{\Delta_2}{3}$.

Similarly, for any sub-optimal arm $i$, we can find $T_i^*(\varepsilon)$ such that with probability $1 - \varepsilon$, $\hat{\mu}_i(t) \leq \mu_i + \frac{\Delta_2}{3}$ for any $t > T_1^*(\varepsilon)$. The only difference in this argument is that we need to use Fact 2 instead of the fact that the algorithm has $o(T)$ regret in expectation.

Let $T^*(\varepsilon) = \max_i T_i^*(\varepsilon)$. We know that after $T^*(\varepsilon)$, with probability at least $1 - N\varepsilon$, pulling arm $i$ once needs at least $\hat{\mu}_1(t) - \hat{\mu}_i(t) \geq \frac{\Delta_i}{3}$ for compensation.

Before time $T^*(\varepsilon)$, every arm can be pulled for at most $T^*(\varepsilon)$ time steps. As $T$ goes to infinity, by Fact 2, every sub-optimal arm $i$ needs to be pulled for at least $(1 - \varepsilon)\frac{\log T}{KL(D_i, D_1)}$ times. Thus the player needs to pay at least $\left((1 - \varepsilon)\frac{\log T}{KL(D_i, D_1)} - T^*(\varepsilon)\right) \times \frac{\Delta_i}{3}$ for compensation on arm $i$ until time $T$.

Taking $T$ going to infinity and setting $\varepsilon = \frac{1}{2N}$, since $T^*(\varepsilon)$ does not depend on $T$, the total compensation is

$$\Omega\left(\sum_{i=2}^{N} \frac{\Delta_i \log T}{KL(D_i, D_1)}\right)$$

## C    Proof of Theorem 2

After the first $N$ time steps, every arm $i$ has $\hat{\mu}_i(t) = M_i(t)/N_i(t)$.

Notice that we always choose the arm $i$ with maximum value $\hat{\mu}_i(t) + r_i(t)$. Thus the arm $a(t)$ satisfies the following inequality:

$$\hat{\mu}_{a(t)}(t) + r_{a(t)}(t) \geq \max_j \hat{\mu}_j(t) + r_j(t) \geq \max_j \hat{\mu}_j(t)$$

This means that we need to pay at most $r_{a(t)}(t)$ for compensation.

For each sub-optimal arm $i \neq 1$, if it is chosen at time $t$, then we must have $\hat{\mu}_i(t) + r_i(t) \geq \hat{\mu}_1(t) + r_1(t)$. Since $\mu_1 = \mu_i + \Delta_i$, we have:

$$\hat{\mu}_i(t) + \mu_1 + 2r_i(t) \geq \mu_i + r_i(t) + \hat{\mu}_1(t) + r_1(t) + \Delta_i$$

This implies that one of the following three events must happen:

$$\mathcal{A}_i^{UCB}(t) = \{\hat{\mu}_i(t) \geq \mu_i + r_i(t)\}$$
$$\mathcal{B}^{UCB}(t) = \{\mu_1 \geq \hat{\mu}_1(t) + r_1(t)\}$$
$$\mathcal{C}_i^{UCB}(t) = \{2r_i(t) \geq \Delta_i\}$$

Thus $\mathbb{E}[N_i(T)] \leq \sum_{t=1}^{T}(\Pr[\mathcal{A}_i^{UCB}(t)] + \Pr[\mathcal{B}^{UCB}(t)] + \Pr[\mathcal{C}_i^{UCB}(t)])$.

Notice that $r_i(t) = \sqrt{\frac{2 \log t}{N_i(t)}}$, then if $N_i(t) > \frac{8 \log T}{\Delta_i^2}$, event $\mathcal{C}_i^{UCB}(t)$ can not happen, which means $\sum_{t=1}^{T}\Pr[\mathcal{C}_i^{UCB}(t)] \leq \frac{8 \log T}{\Delta_i^2}$. As for events $\mathcal{A}_i^{UCB}(t)$ and $\mathcal{B}^{UCB}(t)$, we have the following fact given by Chernoff-Hoeffding's inequality.

**Fact 3.** *For any arm $i$, we have:*

$$\sum_{t=1}^{T}\Pr[\hat{\mu}_i(t) \geq \mu_i + r_i(t)] \leq \frac{1}{t^2}$$

$$\sum_{t=1}^{T}\Pr[\hat{\mu}_i(t) \leq \mu_i - r_i(t)] \leq \frac{1}{t^2}$$

By Fact 3, we have $\sum_{t=1}^{T}(\Pr[\mathcal{A}_i^{UCB}(t)] + \Pr[\mathcal{B}^{UCB}(t)]) \leq \frac{\pi^2}{3}$.

If arm $i$ has been pulled for $N_i(T)$ times, we need to pay compensation for at most $\sum_{k=1}^{N_i(T)} \sqrt{\frac{2 \log T}{k}} \leq \sqrt{8N_i(T) \log T}$, then

$$Com_i(T) \leq \mathbb{E}_{N_i(T)}\left[\sqrt{8N_i(T) \log T}\right] \leq \sqrt{8\mathbb{E}[N_i(T)] \log T} \leq \frac{8 \log T}{\Delta_i} + \frac{\pi^2}{3}.$$

As for arm 1, we can see that when $N_1(t) = \max_i N_i(t)$ and $a(t) = 1$, we do not need to pay compensation. The reason is that $\hat{\mu}_1(t) + r_1(t) \geq \hat{\mu}_i(t) + r_i(t)$ and $r_i(t) \geq r_1(t)$ imply $\hat{\mu}_1(t) \geq \hat{\mu}_i(t)$.

Thus, let $N_1'(T) = \max_{i \neq 1} N_i(T)$, we know that we only need to pay compensation for pulling arm 1 when $N_1(t) \leq N_1'(T)$.

Notice that we have $\mathbb{E}[N_1'(T)] \leq \sum_{i \neq 1} \mathbb{E}[N_i(T)] \leq \sum_{i \neq 1} \frac{8 \log T}{\Delta_i^2} + \frac{N\pi^2}{3}$. Thus, the compensation we need to pay on arm 1 satisfies

$$Com_1(T) \leq \mathbb{E}_{N_1'(T)}\left[\sqrt{8N_1'(T)\log T}\right] \leq \sqrt{8\mathbb{E}[N_1'(T)]\log T} \leq \sum_{i \neq 1} \frac{8\log T}{\Delta_i} + \frac{N\pi^2}{3}.$$

Summing over all sub-optimal arms and the optimal arm, in Algorithm 1, we have

$$Com(T) = \sum_i Com_i(T) \leq \sum_{i=2}^{N} \frac{16 \log T}{\Delta_i} + \frac{2N\pi^2}{3}.$$

## D   Proof for Theorem 3

Notice that only if we choose to explore arm $j$, we need to pay compensation. Now consider the expected compensation we need to pay on exploring arm $j$ for the $k$-th time, which can be written as $\mathbb{E}[\max_i(\hat{\mu}_i(t_j^\varepsilon(k)) - \hat{\mu}_j(t_j^\varepsilon(k)))]$.

Then we can have:

$$
\begin{aligned}
&\mathbb{E}[\max_i(\hat{\mu}_i(t_j^\varepsilon(k)) - \hat{\mu}_j(t_j^\varepsilon(k)))] \\
=\ & \mathbb{E}[\max_i(\hat{\mu}_i(t_j^\varepsilon(k)) - \mu_i + \mu_i - \mu_j + \mu_j - \hat{\mu}_j(t_j^\varepsilon(k)))] && (14) \\
\leq\ & \mathbb{E}[\max_i(\hat{\mu}_i(t_j^\varepsilon(k)) - \mu_i)] + \mathbb{E}[\max_i(\mu_i - \mu_j)] + \mathbb{E}[(\mu_j - \hat{\mu}_j(t_j^\varepsilon(k)))] \\
=\ & \mathbb{E}[\max_i(\hat{\mu}_i(t_j^\varepsilon(k)) - \mu_i)] + \Delta_j + \mathbb{E}[(\mu_j - \hat{\mu}_j(t_j^\varepsilon(k)))] && (15) \\
=\ & \mathbb{E}[\max_i(\hat{\mu}_i(t_j^\varepsilon(k)) - \mu_i)] + \Delta_j && (16)
\end{aligned}
$$

Eq. (15) is because that $\mathbb{E}[\max_i(\mu_i - \mu_j)] = \max_i(\mu_i - \mu_j) = \mu_1 - \mu_j = \Delta_j$, and Eq. (16) is because that whether we choose to explore arm $j$ are independent with the its observations.

Now we consider the value $\mathbb{E}[\max_i(\hat{\mu}_i(t) - \mu_i)]$. It is upper bounded by $\sum_i \mathbb{E}[(\hat{\mu}_i(t) - \mu_i)^+]$. When $j$ is explored for $k$ times, we know every arm must have be chosen for at least $k$ times. Notice that these feedbacks are independent with whether we choose to explore or not. Thus, we have:

$$
\begin{aligned}
\sum_i \mathbb{E}[(\hat{\mu}_i(t_j^\varepsilon(k)) - \mu_i)^+] &= \frac{1}{2}\sum_i \mathbb{E}[|\hat{\mu}_i(t_j^\varepsilon(k)) - \mu_i|] \\
&\leq \frac{1}{2}\sum_i \sqrt{\mathbb{E}[(\hat{\mu}_i(t_j^\varepsilon(k)) - \mu_i)^2]} \\
&\leq \frac{1}{2}\sum_i \sqrt{\frac{1}{4k}} \\
&= \frac{N}{4\sqrt{k}}
\end{aligned}
$$

Suppose arm $j$ has been explored for $n_j(T)$ times until time $T$. Then

$$\sum_{k=1}^{n_j(T)} \mathbb{E}[\max_i(\hat{\mu}_i(t_j^\varepsilon(k)) - \hat{\mu}_j(t_j^\varepsilon(k)))] \leq \sum_{k=1}^{n_j(T)}\left(\frac{N}{4\sqrt{k}} + \Delta_j\right) \leq n_j(T)\Delta_j + \frac{N}{2}\sqrt{n_j(T)}$$

Notice that when $\epsilon = \frac{cN}{\Delta_2^2}$, $\mathbb{E}[n_j(T)] = \frac{c \log T}{\Delta_2^2}$, then $\mathbb{E}[n_j(T)\Delta_j + \frac{N}{2}\sqrt{n_j(T)}] \leq \frac{c\Delta_j \log T}{\Delta_2^2} + \frac{N}{2\Delta_2}\sqrt{c \log T}$. Thus the total compensation is upper bounded by:

$$\sum_{i=2}^{N} \frac{c\Delta_i \log T}{\Delta_2^2} + \frac{N^2}{2\Delta_2}\sqrt{c \log T}$$

## E    Proof for Theorem 4

We first give four important lemmas, which come from the analysis of TS policy in previous works [1, 11]. Their proofs can be modified slightly to work in our Algorithm 3.

**Lemma 4.** *(Theorem 1 in [1]) In Algorithm 3, summing over all possible rounds $(t, t+1)$, we have that for all $i \neq 1$ and $\varepsilon < \Delta_i$:*

$$\sum_{(t,t+1)} \Pr[a_2(t+1) = i] = \frac{2}{(\Delta_i - \varepsilon)^2} \log T + O\left(\frac{1}{\varepsilon^4}\right)$$

**Lemma 5.** *(Proposition 1 in [11]) In Algorithm 3, summing over all possible rounds $(t, t+1)$, we have that*

$$\sum_{(t,t+1)} \Pr[N_1(t) \leq t^b] \leq C(\boldsymbol{\mu})$$

*holds for some constant $b = b(\boldsymbol{\mu}) \in (0,1)$.*

**Lemma 6.** *(Lemma 2 in [1]) In Algorithm 3, summing over all possible rounds $(t, t+1)$, for any $i \neq 1$, we have that*

$$\sum_{(t,t+1)} \Pr[a_1(t) = i, \hat{\mu}_i(t) \geq \mu_i + \frac{\Delta_i}{2}] \leq \frac{4}{\Delta_i^2} + 1$$

**Lemma 7.** *In Algorithm 3,*

$$\forall i, \Pr[|\theta_i(t) - \hat{\mu}_i(t)| \geq r_i(t)] \leq \frac{1}{t^2}$$

*where $r_i(t) = \sqrt{\frac{2 \log t}{N_i(t)}}$.*

We do not provide proofs for Lemma 4, Lemma 5 and Lemma 6 since they are almost the same as in [1] and [11]. As for Lemma 7, although the proof is similar, the statement is not the same. Thus we provide its proof in the end of this section.

Firstly, we analyze the regret bound of Algorithm 3.

Lemma 4 shows that the regret during sample steps are bounded, now we come to the regret during empirical steps.

We use the following four events to help our analysis:

$$
\begin{aligned}
\mathcal{A}_i^{TS}(t) &= \{\hat{\mu}_i(t) \geq \mu_i + \frac{\Delta_i}{2}\}\\
\mathcal{B}^{TS}(t) &= \{\hat{\mu}_1(t) + r_1(t) \leq \mu_1\}\\
\mathcal{C}_i^{TS}(t) &= \{2r_1(t) \geq \Delta_i\}\\
\mathcal{D}^{TS}(t) &= \{N_1(t) > t^b\}
\end{aligned}
$$

Then:

$$
\begin{aligned}
\sum_{(t,t+1)} \Pr[a_1(t) = i] \leq{}& \sum_{(t,t+1)} \Pr[\mathcal{A}_i^{TS}(t) \cap \{a_1(t) = i\}] + \sum_{(t,t+1)} \Pr[\mathcal{B}^{TS}(t) \cap \{a_1(t) = i\}]\\
&+ \sum_{(t,t+1)} \Pr[\mathcal{C}_i^{TS}(t) \cap \mathcal{D}^{TS}(t) \cap \{a_1(t) = i\}] + \sum_{(t,t+1)} \Pr[\neg\mathcal{D}^{TS}(t) \cap \{a_1(t) = i\}]
\end{aligned}
$$

$$+ \sum_{(t,t+1)} \Pr[\neg\mathcal{A}_i^{TS}(t) \cap \neg\mathcal{B}^{TS}(t) \cap \neg\mathcal{C}_i^{TS}(t) \cap \{a_1(t) = i\}])$$

Lemma 6 shows that $\sum_{(t,t+1)} \Pr[\mathcal{A}_i^{TS}(t) \cap \{a_1(t) = i\}] \leq \frac{4}{\Delta_i^2} + 1$. Using Fact 3, $\sum_{(t,t+1)} \Pr[\mathcal{B}^{TS}(t) \cap \{a_1(t) = i\}] \leq \sum_{(t,t+1)} \Pr[\mathcal{B}^{TS}(t)] \leq \frac{\pi^2}{6}$.

Then we consider $t$ such that $\mathcal{C}_i^{TS}(t) \cap \mathcal{D}^{TS}(t)$ happens. By definition, we can see that $\frac{\Delta_i}{2} \leq r_1(t) = \sqrt{\frac{2\log t}{N_1(t)}} \leq \sqrt{\frac{2\log t}{t^b}}$. Thus there exists $t_i = f(i, \boldsymbol{\mu})$ such that for all $t \geq t_i$, $\Pr[\mathcal{C}_i^{TS}(t) \cap \mathcal{D}^{TS}(t)] = 0$. This implies that:

$$\sum_{(t,t+1)} \Pr[\mathcal{C}_i^{TS}(t) \cap \mathcal{D}^{TS}(t) \cap \{a_1(t) = i\}] \leq \sum_{(t,t+1)} \Pr[\mathcal{C}_i^{TS}(t) \cap \mathcal{D}^{TS}(t)] \leq t_i$$

$\sum_{(t,t+1)} \Pr[\neg\mathcal{D}^{TS}(t) \cap \{a_1(t) = i\}]$ is upper bounded by Lemma 5, which is $C(\boldsymbol{\mu})$.

$\neg\mathcal{A}_i^{TS}(t) \cap \neg\mathcal{B}^{TS}(t) \cap \neg\mathcal{C}_i^{TS}(t) \cap \{a_1(t) = i\}$ cannot happen since under the first three events we have:

$$\hat{\mu}_1(t) > \mu_1 - r_1(t) > \mu_1 - \frac{\Delta_i}{2} = \mu_i + \frac{\Delta_i}{2} > \hat{\mu}_i(t),$$

which contradict with $\{a_1(t) = i\}$.

Thus, we have that $\sum_{i=2}^{N} \left( \sum_{(t,t+1)} \Pr[a_1(t) = i] \right) \leq N \left( 1 + \frac{\pi^2}{6} \right) + \sum_{i=2}^{N} \left( \frac{4}{\Delta_i^2} + t_i \right) + C(\boldsymbol{\mu}) = F_1(\boldsymbol{\mu})$ for some function $F_1$, and it is independent with time horizon $T$.

Adding the regret during sample steps, the total regret of Algorithm 3 is upper bounded by $\sum_i \frac{2\Delta_i}{(\Delta_i - \varepsilon)^2} \log T + O\left(\frac{1}{\varepsilon^4}\right) + F_1(\boldsymbol{\mu})$.

Now we consider the compensation. Notice that in empirical steps we always choose the arm with the largest empirical mean, thus we do not need to pay any compensation in this time slot. Because of this, we can focus on the compensation in sample steps. To do so, we define a event $\mathcal{E}^{TS}(t)$ as following:

$$\mathcal{E}^{TS}(t) = \{\forall i, |\theta_i(t) - \hat{\mu}_i(t)| \leq r_i(t)\}$$

Then the total compensation can be written as

$$\mathbb{E}[\sum_{(t,t+1):t+1<T} c_i(t+1)] \leq \mathbb{E}[\sum_{(t,t+1):t+1<T} \mathbb{I}[\mathcal{E}^{TS}(t)]c_i(t+1)] + \mathbb{E}[\sum_{(t,t+1):t+1<T} \mathbb{I}[\neg\mathcal{E}^{TS}(t)]c_i(t+1)]$$

$$(17)$$

Lemma 7 shows that $\mathbb{E}[\sum_{(t,t+1):t+1<T} \mathbb{I}[\neg\mathcal{E}^{TS}(t)]] \leq \frac{N\pi^2}{6}$, thus the second term in Eq. (17) has upper bound $\frac{N\pi^2}{6}$ as well.

Now we consider the first term in Eq. (17). Here the compensation we need to pay is $c_i(t+1) = \max_j \hat{\mu}_j(t+1) - \hat{\mu}_{a_2(t+1)}(t+1)$.

If $a_1(t) = a_2(t+1) = i$ and $\hat{\mu}_i(t+1) \geq \hat{\mu}_i(t)$, we have $c_i(t+1) = 0 < \frac{1}{N_i(t)}$.

If $a_1(t) = a_2(t+1) = i$ but $\hat{\mu}_i(t+1) < \hat{\mu}_i(t)$, then we know that $\max_j \hat{\mu}_j(t+1) \leq \hat{\mu}_i(t)$, thus $c_i(t+1) \leq \hat{\mu}_i(t) - \hat{\mu}_i(t+1) \leq \frac{1}{N_i(t)}$.

If $a_1(t) = k, a_2(t+1) = i$ and $k \neq i$, then $c_i(t+1) = \max_j \hat{\mu}_j(t+1) - \hat{\mu}_i(t+1) \leq \hat{\mu}_k(t) + \frac{1}{N_k(t)} - \hat{\mu}_i(t)$.

Thus, if $a_1(t) = a_2(t+1)$, we need to pay at most $\frac{1}{N_i(t)}$ for compensation. Otherwise, we need to pay at most $(\hat{\mu}_{a_1(t)}(t) - \hat{\mu}_{a_2(t+1)}(t)) + \frac{1}{N_{a_1(t)}(t)}$.

Notice that under event $\mathcal{E}(t)$, $\hat{\mu}_{a_2(t+1)}(t) + r_{a_2(t+1)}(t) \geq \theta_{a_2(t+1)}(t) \geq \theta_{a_1(t)}(t) \geq \hat{\mu}_{a_1(t)}(t) - r_{a_1(t)}(t)$. Thus if $a_1(t) \neq a_2(t+1)$, $c_i(t+1) \leq r_{a_1(t)}(t) + r_{a_2(t+1)}(t) + \frac{1}{N_{a_1(t)}(t)}$. Then we can treat the total compensation as following: we first pay $r_{a_1(t}(t) + \frac{1}{N_{a_1(t)}(t)}$ on arm $a_1(t)$ in the empirical step, and then pay $r_{a_2(t+1)}(t)$ on arm $a_2(t+1)$ in the sample step. By this method, we can upper bound the compensation we need to pay on pulling sub-optimal arm $i$ as $\sum_{\tau=1}^{N_i(T)} \sqrt{\frac{2 \log T}{\tau}} + \frac{1}{\tau} \leq \log T + \sqrt{8 N_i(T) \log T}$ under the event $\mathcal{E}^{TS}(t)$.

By regret analysis, $\mathbb{E}[N_i(T)] \leq \frac{2}{(\Delta_i - \varepsilon)^2} \log T + O\left(\frac{1}{\varepsilon^4}\right) + t_i + \frac{4}{\Delta_i^2} + \frac{\pi^2 + 6}{6} + C(\boldsymbol{\mu})$, thus the compensation $Com_i(T)$ is upper bounded by $\frac{4}{\Delta_i - \varepsilon} \log T + O\left(\frac{1}{\varepsilon^4}\right) + t_i + \frac{4}{\Delta_i^2} + \frac{\pi^2 + 6}{6} + C(\boldsymbol{\mu}) + \log T$.

As for arm 1, when $a_1(t) = a_2(t+1) = 1$, we only need to pay $\frac{1}{N_1(t)}$ for compensation. The expected number of time steps that we do not have $a_1(t) = a_2(t) = 1$ is at most $\sum_{i=2}^{N} \frac{2}{(\Delta_i - \varepsilon)^2} \log T + F(\boldsymbol{\mu}) + O\left(\frac{N}{\varepsilon^4}\right)$, which is given by the regret analysis. This means that the compensation on pulling arm 1 is upper bounded by $\sum_{i=2}^{N} \frac{4}{\Delta_i - \varepsilon} \log T + F_1(\boldsymbol{\mu}) + O\left(\frac{N}{\varepsilon^4}\right) + \log T$.

Thus, the first term in Eq. (17) has upper bound $\sum_{i=2}^{N} \frac{8}{\Delta_i - \varepsilon} \log T + F_1(\boldsymbol{\mu}) + O\left(\frac{N}{\varepsilon^4}\right) + N \log T + \sum_{i=2}^{N} \left(t_i + \frac{4}{\Delta_i^2} + \frac{\pi^2 + 6}{6} + C(\boldsymbol{\mu})\right)$. After adding the upper bound $\frac{N\pi^2}{6}$ of the second term and setting $F_2(\boldsymbol{\mu}) = F_1(\boldsymbol{\mu}) + \frac{N\pi^2}{6} + \sum_{i=2}^{N} \left(t_i + \frac{4}{\Delta_i^2} + \frac{\pi^2 + 6}{6} + C(\boldsymbol{\mu})\right)$, we have that

$$Com(T) \leq \sum_i \frac{8}{\Delta_i - \varepsilon} \log T + N \log T + O\left(\frac{N}{\varepsilon^4}\right) + F_2(\boldsymbol{\mu}).$$

### E.1 Proof of Lemma 7

Since $\theta_i(t)$ only depends on the values of $(\alpha_i(t), \beta_i(t))$ but is independent of the random history, we can fix the pair $(\alpha_i(t), \beta_i(t))$ to prove the inequality, and then the inequalities hold also for $(\alpha_i(t), \beta_i(t))$ as random variables.

$$
\begin{aligned}
\Pr[\theta_i(t) > \hat{\mu}_i(t) + r_i(t)] &= 1 - F^{Beta}_{\alpha_i(t), \beta_i(t)(\hat{\mu}_i(t) + r_i(t))} \\
&= 1 - (1 - F^{B}_{\alpha_i(t) + \beta_i(t) - 1, \hat{\mu}_i(t) + r_i(t)}(\alpha_i(t) - 1)) \quad (18) \\
&= F^{B}_{\alpha_i(t) + \beta_i(t) - 1, \hat{\mu}_i(t) + r_i(t)}(\alpha_i(t) - 1) \\
&\leq F^{B}_{\alpha_i(t) + \beta_i(t) - 1, \hat{\mu}_i(t) + r_i(t)}(\hat{\mu}_i(t)(\alpha_i(t) + \beta_i(t) - 1)) \\
&\leq \exp(-(\alpha_i(t) + \beta_i(t) - 1) KL(\hat{\mu}_i(t), \hat{\mu}_i(t) + r_i(t))) \quad (19) \\
&\leq \exp(-\frac{N_i(t) r_i(t)^2}{2}) \quad (20) \\
&\leq \frac{1}{t^2}
\end{aligned}
$$

Eq. (18) is given by the following Beta-Binomial Trick (Fact 4), Eq. (19) is given by Chernoff-Hoeffding Inequality, and Eq. (20) follows the fact that $KL(x, y) \geq \frac{|x-y|^2}{2}$.

**Fact 4.** *(Beta-Binomial Trick) Let $F^{Beta}_{a,b}(x)$ be the cdf of Beta distribution with parameters $(a, b)$, let $F^{B}_{n,p}(x)$ be the cdf of Binomial distribution with parameters $(n, p)$. Then for any positive integers $(a, b)$, we have*

$$F^{Beta}_{a,b}(x) = 1 - F^{B}_{a+b-1, x}(a - 1)$$