[Reviews · NeurIPS 2018]

Reviewer 1



This paper studies a variant of the classical stochastic multi-armed bandit problem they call the "known-compensation multi-arm bandit" problem (KCMAB), which serves as a model for the problem of incentivizing exploration. In this model, the principal (i.e. the system controller) cannot pull an arm directly; instead, each round a short-term (/myopic) agent arrives, and the principal can try to convince this agent (by offering some coompensation depending on which arm the agent picks) to pick an arm of the principal's choice. The agents see all previous rewards, so in the absence of any compensation they will each pick the arm with the largest historical average reward so far, essentially executing Follow-The-Leader (which is known to have regret linear in T). The principal can convince an agent to play some other arm i by offering compensation (if the agent picks that arm) equal to the difference between the historical average of arm i and the largest historical average of any arm. The authors show two matching lower and upper bounds for the best compensation possible while optimizing regret (up to constant factors). The lower bound shows that there is some inherent tradeoff between regret and compensation: if a strategy for the principal achieves sub-polynomial regret (o(T^alpha) for any alpha > 0), then the principal must pay at least Omega(log T) in compensation. The upper bound shows that UCB (with appropriate compensation so that arms play according to UCB) essentially achieves this lower bound, achieving both O(log T) regret and compensation. (The actual statements are a bit more precise, with the explicit dependence of both regret/compensation on the distributions/means of the arms). The paper also studies two other algorithms, the epsilon-greedy algorithm and a variant of Thompson sampling, and show that both achieve O(log T) regret/compensation. In addition, they run some simple experiments for all of these algorithms on synthetic data, and show that they match up with the theoretical results. I think this is an interesting paper. It is true that very similar notions of "compensation" have been studied before in the incentivizing exploration literature -- Frazier-Kempe-Kleinberg-Kleinberg in EC 2014 essentially study this quantity in the case of an unbounded time horizon for a principal with discounted utility, and their setting also displays many of the phenomena discovered in this paper (in fact in their model they can get a much more precise version of the tradeoff between regret and compensation). That said, I'm surprised that (to the best of my knowledge) this specific notion of "compensation of a bandit algorithm" (for a finite time horizon) has not been explicitly studied, and I think it is an interesting quantity to understand. I think there are also a number of interesting follow up questions that can be asked (e.g. can we say anything about compensation required to implement adversarial bandit algorithms like EXP3?). In some sense, the upper/lower bounds are not too surprising; both regret and compensation come from "exploring" (whereas "exploiting" does not cost any compensation, and only costs regret if you have not explored sufficiently). Of course, this intuition is far from a proof. The proof of Theorem 1 (the main lower bound) is quite nice, and is probably the most interesting technical result of the paper. The upper bounds generally follow from the analysis of the corresponding bandit algorithms and examining when/how much compensation is required. This paper is well-written and easy to read. I recommend this paper for acceptance.

Reviewer 2



This paper is about multi-armed bandits. The twist is that the learner does not control directly the actions. Instead, in each round the learner chooses an amount of 'compensation' associated to each arm and the user chooses the arm for which the sum of the compensation and empirical mean is largest. The objective is to design algorithms for which the cumulative regret (with the normal definition) and cumulative compensation both have logarithmic growth. The claimed results are as follows: (a) A lower bound showing that algorithms with subpolynomial regret necessarily must necessarily pay logarithmic cumulative compensation with the usual constants on *some* bandit problems. (b) Upper bounds on the cumulative compensation for UCB, epsilon-greedy and a variant of Thompson sampling with appropriately chosen compensation levels. To illustrate, the most obvious idea of offering the difference between the UCB and empirical mean as compensation leads to a sqrt(n) growth rate of the compensation, so something a little clever is needed. The authors point out the compensation is not required if the empirical mean of the arm chosen by UCB is the largest and then prove this happens with sufficiently few exceptions. By the way, one might wonder if the DMED-style algorithm (Honda and Takamura, 2010) might not be adapted even more straightforwardly. I have two reservations about this paper. First, the proofs are much more complicated than necessary. This is especially true of the lower bound, the proof of which is three pages. I include a suggestion for how to considerably shorten this proof below. Note that this sketch does not depend on the first arm having a mean of at least 0.9 and applies to all bandit problems, like Lai & Robbins result. My second reservation is that the paper is not very cleanly written. There are many typos and the sketches are hard to follow. The proofs in the appendix even harder. And yet the claimed results do not seem especially surprising. I think a paper like this could be accepted at NIPS, but the analysis should be crisp and insightful. Regarding the originality and significance. The setting is new to my knowledge. The techniques are not especially startling. The significance is hard to judge. There is some motivation given, but this is obviously a first step. In practical situations things would almost always be contextualized and so-on. Still, I think this aspect is Ok. A few other comments: 1. I was wondering if perhaps you can prove a more generic result. Given /any/ algorithm can I design a black-box modification for which the compensation bound has the same order of the regret for that algorithm? 2. Perhaps you can get the constants right in the lower bound and upper bound. See below for the former. For the latter you could use the techniques from the KL-UCB algorithm. 3. All of the upper bounds involve simple modification of existing algorithms. Essentially what is required is choosing the compensation so that the behavior is the same as what you would expect in the standard setting. The lower bounds show you cannot do much better than this, but one still might wonder if the algorithms ought to change a little in the new setting. Mathematically it seems a pity the setting is not especially rich. Overall the paper is below the bar for NIPS. ********************************************** * LOWER BOUND SKETCH ********************************************** By Lai & Robbins result it holds that P(N_i(T) >= (1-e) / KL log(n)) >= 1 - delta(T) with delta(T) tending to 0. Let epsilon > 0. I claim there exists a constant m that depends on the bandit, but not T such that the following four events hold jointly with probability at least 1/2 for all sufficiently large T: (a) N_i(T) >= (1 - e) / KL log(T) for all i (b) N_1(2m) >= m (c) Whenever N_1(t) >= m, then the empirical mean of arm 1 is at least mu_1 - epsilon (d) Whenever N_i(t) >= m, then the empirical mean of arm i is a most mu_i + epsilon for all i Then since m does not depend on T we may take the limit as T tends to infinity when on the four events above it the algorithm must pay at least Delta_i - 2 epsilon for at least (1 - e) / KL log(T) - m rounds, which yields the desired form of the lower bound. The claim is proven easily via a union bound. (a) using Lai & Robbins result, (b) by Markov's inequality and the assumption on subpolynomial regret. (c) and (d) by Hoeffding's bound and a union bound (Chernoff's is not required). *********************** * MINORS *********************** * The inequality in the sketch of Theorem 1 is the wrong way. * \Theta(f(n)) means that f(n) = O(f(n)) and f(n) = Omega(f(n)), so the claim of Theorem 1 is not accurate. In general I prefer not to see this notation in theorem statements.

Reviewer 3



This work is regarding multi-armed bandits problem where the controller is able to set compensation in order to guide greedy players out of seemingly optimal alternatives. The paper itself is well written with sound structure and clear demonstration. In particular, this paper proposed a "known-compensation MAB" setting, which is interesting in a theoretical point of view as well as for application scenario. In addition to some bounds proved for the newly proposed setting, the authors also demonstrate the counterparts of the existing MAB policies for the proposed KCMAB scenario including epsilon-greedy, Thompson sampling, and UCB. I think this work is innovative and good enough to be considered publication in NIPS.